# *MyoDex*: Generalizable Representations for Dexterous Physiological Manipulation

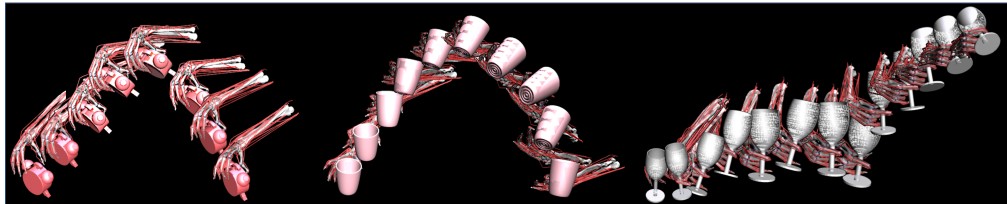

**Figure 1:** Contact rich manipulation behaviors acquired by *MyoDex* with physiological *MyoHand*

## Abstract

The complexity of human dexterity builds on the coordinated actuation of a large number of muscles. Still, much is to be understood about how the control of such overactuated system for hand manipulation behaviors emerge and quickly and flexibly adapts to new behaviours. In this work we aim at learning generalizable representations for dexterous manipulation behaviors with a physiologically realistic hand model: *MyoHand*. In contrast to prior works demonstrating isolated postural and force control, here we demonstrate musculoskeletal agents (*MyoDex*) exhibiting contact-rich dynamic dexterous manipulation behaviors in simulation. Furthermore, to demonstrate generalization, we show that a single *MyoDex* agent can be trained to solve up-to 14 different contact-rich tasks. Aligned with human development, simultaneous learning of multiple tasks imparts physiological coordinated muscle contractions i.e., muscle synergies, that are not only shared amongst those in-domain tasks but are also effective to a large series of new out-of-domain tasks. By leveraging these pre-trained manipulation synergies, we show generalization to 38 additional previously unsolved tasks. While physiological behaviors with large muscle groups (such as legged-locomotion, arm-reaching, etc) have been demonstrated before, to the best of our knowledge nimble behaviors of this complexity with smaller muscle groups and generalizable representations for the control of the overactuated human hand are being demonstrated for the first time.

**Project Webpage**: https://sites.google.com/view/myodex

Musculoskeletal, muscle synergies, Machine Learning, human dexterity

## 1 Introduction

Human hands are astonishingly complex and require effective coordination of various muscle groups to impart effective manipulation abilities. Manipulation behaviors are incredibly sophisticated as, because of the overactuated musculoskeletal system, they evolve in a high-dimensional search space populated with intermittent contact dynamics between the hands' degrees of freedom and the object. Indeed, even in the field of robotics where joints and actuations are simpler, finding effective manipulation strategies nonetheless remains a challenge Kumar et al. (2016); Rajeswaran et al. (2018); Nagabandi et al. (2020).

The human hand consists of 29 bones, 23 joints, and more than 50 muscles Sobinov & Bensmaia (2021). The complex multi-articular, multi-joint, pulling-only properties of the musculoskeletal system Sobinov & Bensmaia (2021) make physiological dexterous manipulation a very different and unique problem as opposed to joint based control typically adopted in robotics. In biology, the control of such complex musculoskeletal system is made possible by the fact that muscles are

not activated in isolation, but rather, that different muscles are activated in a proportional way as a unit. This phenomenon is known as muscle synergy Bizzi & Cheung (2013). Synergies allows the biological motor system – via the modular organization of the movements in the spinal cord Bizzi & Cheung (2013); Caggiano et al. (2016) – to simplify the control problem, solving tasks by building on a limited number of shared solutions d'Avella et al. (2003); d'Avella & Bizzi (2005). Those shared synergies are suggested to be the fundamental building blocks for quickly learning new and more complex motor behaviours Yang et al. (2019); Dominici et al. (2011); Cheung et al. (2020). Manipulation behaviors, the subject of this investigation, are further complicated because they unfold on a sequence of phases: reaching to the object, hand-object contact, and manipulation with object maneuvers. Before the hand-object contact, the human hand is pre-shaped to conform to the object such that it is often possible to predict the object that is going to be grasped just by observing the hand pose before hand-object contact Jeannerod (1988); Santello et al. (2002); Thakur et al. (2008); Yan et al. (2020). Contact and manipulation of the object are goal-driven so that the way the object is held depends on both the object affordance and the intermediate task goals Jeannerod (1988).

In this work, we seek to further our understanding of physiological dexterity by imparting dexterous manipulation ability to an anatomically realistic hand-fore-arm model Caggiano et al. (2022). While prior works have not been able to scale beyond dexterous grasping McFarland et al. (2021); Mirakhorlo et al. (2018); Saito et al. (2021); Crouch & Huang (2015); Engelhardt et al. (2021) in a controlled setting with a physiologically realistic models of the hand, here we present *MyoDex* agents capable of dynamic dexterous contact rich manipulation behaviors with multiple objects and a variety of tasks e.g. drinking from a cup, playing with toys, etc. Furthermore, by jointly training multiple tasks, we capture reusable synergies in form of a general pre-trained policy that can be further fine-tuned to manipulate 38 previously unsolved tasks with non-trivial affordances. We provide a detailed analysis of emergent physiological details in our achieved behaviors.

While we do not claim to have solved physiological dexterous manipulation, we emphasize that manipulation abilities demonstrated here significantly advance the state of the art of the bio-mechanics and neuroscience fields. Along these lines, this investigation is among the first to yield robust control policies exhibiting basic physiological constructs such as synergistic activations of muscle groups during dexterous manipulations. Nevertheless, further work is required to rigorously ground them in experimental validation. More specifically, our main contributions are:

- We show for the first time that despite the high numbers of degrees of freedom, the multi-articular-multi-joint and the third order muscle dynamics of muscle control, **it is possible to control a physiologically realistic musculoskeletal model of the hand to perform contact-rich skilled manipulation** behaviors on up-to 14 different tasks.

- We show that joint multi-task learning facilitates the **learning of physiological representations that exploit muscle coordination in a lower-dimensional space of synergies** to solve specific tasks.

- Our framework *MyoDex* **leverages joint multi-task learning to recover reusable representations (synergies) that allows for easier fine-tuning in both in-domain and out-of-domain tasks** (including one/few shot learning). Leveraging these synergies the *MyoDex* solves 38 previously unsolved tasks.

## 2 RELATED WORKS

Experimental studies of functional hand manipulations have been limited both by challenges in sensing, the discontinuous hand-object interactions and because of the limited ability to record many muscles of the hand simultaneously. Musculoskeletal models of the hand McFarland et al. (2021); Lee et al. (2015); Saul et al. (2015) have been developed to overcome some of the experimental limitations and produce insights on the kinematic information of the muscles and joints. While musculoskelatal models of large muscle groups have been extensively developed and used Delp et al. (2007); Seth et al. (2018), models of the hand have been more challenging both because of the smaller muscle groups involved and the complexity of the behaviour they can produce. Indeed, simulations of the hand mostly focus on fingertips, pinch force McFarland et al. (2021), kinematic motion McFarland et al. (2021), and passive grasping McFarland et al. (2021). Furthermore, most of those studies are also limited by intensive computational needs and restricted contact forces. Those

conditions prevented the study of complex hand object interactions and limited by using optimization methods that could not leverage data-driven state of the art.

Recently, a new hand and wrist model – MyoHand Caggiano et al. (2022); Wang et al. (2022) – has been developed. This model overcomes some limitations of alternative hand models and it is suitable for computationally intensive data-driven explorations. Indeed, it has been shown that MyoHand can be trained to solve individual in-hand tasks on very simple geometries (ball, pen).

Hand dexterity has been also a very active field in robotics. Robotic hands have been shown to be able to perform complex in-hand manipulation of real-world objects OpenAI et al. (2019); Huang et al. (2021); Chen et al. (2021) and solve complex manipulation tasks, such as *HandManipulateEgg* and *HandManipulatePen* Plappert et al. (2018). Still, both the hardware and the control of robotic hands do not match the level of dexterity of human hands and remain limited to in-hand movements.

Data driven approaches have consistently used Reinforcement Learning (RL) on joint-based control to solve simple locomotion tasks Miki et al. (2022), in animation of physics based characters Heess et al. (2017) and to solve complex dexterous manipulation in robotics Rajeswaran et al. (2018); Kumar et al. (2016); Nagabandi et al. (2019); Chen et al. (2021). Typically, in order to yield more naturalistic movements, different methods have leveraged motion capture data Merel et al. (2017; 2019); Hasenclever et al. (2020). By means of those approaches, it has been possible to learn complex movements and athletic skills such as high jumps Yin et al. (2021), boxing and fencing Won et al. (2021) or playing basketball Liu & Hodgins (2018). More recently, approaches that prime models with hand pre-shaped for a specific task have been shown to be successful at simplifying the search of RL solutions on complex robotic manipulations Dasari et al. (2022). In contrast to joint-based control, in biomechanical models machine learning has been applied on muscle actuators to control movements and produce more naturalistic behaviors. This is a fundamentally different problem than robotic control as the overactuated control space of biomechanical systems leads to ineffective explorations Schumacher et al. (2022). Wang et al. Wang et al. (2012), Geijtenbeek et al. Geijtenbeek et al. (2013), Borno et al. 2020 Al Borno et al. (2020), and Ruckert et al. Rückert & d'Avella (2013), have been using optimization methods on biomechanical models to synthesize walking and running, reaching movements, and biped locomotion. More recently, deep reinforcement learning has been used to either map the muscle-actuation to joint-actuation control to produce movements that are more human-like than those generated by torque-based control at the joints Jiang et al. (2019), in order to directly control shoulder and arm muscles for isometric arm movements Joos et al. (2020) and reaching Schumacher et al. (2022); Ikkala et al. (2022), hand muscles for hand dexterous manipulations Caggiano et al. (2022), co-learning elbow exoskeleton movements Caggiano et al. (2022); Wang et al. (2022), walking/running Song et al. (2020); Park et al. (2022), or to produce movements such as juggling, weight lifting, cart-wheeling and other highly stylistic movements Lee et al. (2018; 2019). Musculoskeletal models have been used also to improve the realism of simulated animal movements; for example, in controlling movements in animal models of a dog Stark et al. (2021) and more recently, of an ostrich Barbera et al. (2021); Schumacher et al. (2022).

While musculoskeletal control with large muscles groups have been demonstrated Song et al. (2020; 2021); Schumacher et al. (2022); Ikkala et al. (2022), nimble contact rich musculoskeletal behaviors with smaller muscle groups such as hand-manipulation remains an open challengeCaggiano et al. (2022). *MyoDex*, in addition of showing that indeed it is possible, presents evidence that the learned physiological representations share muscle coordination across tasks which, like human synergistic control, facilitate both learning and generalization across tasks.

## 3 Physiological Dexterity

Human hand dexterity builds on the fundamental characteristics of the physiological actuation: muscle are multi-articular and multi-joints, the dynamics of the muscle is of the third order, muscle have pulling only capabilities, and coordinated synergistic muscle control with intermittent contact with objects. Furthermore, the key aspect of the control of such physiological effectors is that the human central nervous system optimizes movements through coordinated muscle contraction – muscle synergies – which are meant to simplify the control problem, allowing generalization. Fields of bio-mechanics, rehabilitation, neuro-surgery, etc. have long benefited from physiological understanding of neuro-mechanical control. To further our understanding, here we embed the same

control challenges e.g. by controlling a physiologically accurate musculoskeletal model of the hand (see Sec. 3.1) in complex manipulation tasks (see Sec. 3.2). This allows a window to peek into the mechanisms behind human dexterity that enables generalization across different tasks.

## 3.1 Physiologically accurate MyoHand

In order to simulate a physiologically accurate hand model, a complex musculoskeletal hand model comprised of 29 bones, 23 joints, and 39 muscles-tendon units Wang et al. (2022) - MyoHand model - implemented in the MuJoCo physics simulator Todorov et al. (2012) was used (see Figure A.1). This hand model has been shown to allow dexterous in-hand manipulation of one or multiple objects when trained using reinforcement learning Caggiano et al. (2022).

We extended the MyoHand model to include translations and rotations at the level of the shoulder. We limited the translation on the frontal (range between $[-0.07, \ 0.03]$) and longitudinal (range between $[-0.05, \ 0.05]$) axis to favor shoulder and wrist rotation.

## 3.2 Task

Dexterous manipulation is often posed as a problem of achieving the final configuration of the object. In this study we are interested to capture the whole continuous aspects of the manipulation behaviour with object maneuver e.g., drinking, playing, or cyclic movement like hammering. Those tasks are hard to capture as goal reaching. To effectively capture the temporal behaviors, we instead define dexterous manipulation as a task of realizing a desired object trajectory ($\hat{X}$). We use two metrics to measure task performance. The object error metric $E(\hat{X})$ calculates the average Euclidean distance between the object's center-of-mass position[1], and the desired position from the desired trajectory: $E(\hat{X}) = \frac{1}{T} \sum_{t=0}^{T} \|x_t^p - \hat{x}_t^p\|_2$. In addition, the success metric $S(\hat{X})$ reports the fraction of time-steps where object error is below a $\epsilon = 1cm$ threshold. It is defined as: $S(\hat{X}) = \frac{1}{T} \sum_{t=0}^{T} \mathbb{1} \|x_t^p - \hat{x}_t^p\|_2 < \epsilon$

## 3.3 Dexterity objectives

While the complexity of human level dexterity can be hard to fully quantify, none the less we outline a few objective measures we consider in this work. First, a dexterous agent should be capable of exhibiting contact-rich manipulation behaviors. Next, the agent's behavior should seamlessly generalize to multiple tasks/objects without additional assumptions. Finally, these agents should exhibit coordinated muscles movements (synergies) that are shared amongst different behaviors, as well as generalize to new unseen tasks.

## 4 *MyoDex*: Acquiring dexterity

In this section we discuss our approach to build agents that can learn contact-rich manipulation behaviors and generalize across tasks.

## 4.1 Problem formulation

A manipulation task can be formulated as a Markov Decisions Process (MDP) Sutton & Barto (2018) and solved via Reinforcement Learning (RL). In RL paradigms, the Markov decision process is defined as a tuple $\mathcal{M} = (\mathcal{S}, \mathcal{A}, \mathcal{T}, \mathcal{R}, \rho, \gamma)$, where $\mathcal{S} \subseteq \mathbb{R}^n$ and $\mathcal{A} \subseteq \mathbb{R}^m$ represent the continuous state and action spaces respectively. The unknown transition dynamics is described by $s' \sim \mathcal{T}(\cdot|s,a)$. $\mathcal{R} : \mathcal{S} \to [0, R_{\max}]$, denotes the reward function, $\gamma \in [0,1)$ denotes the discount factor, and and $\rho$ the initial state distribution. In RL, a policy is a mapping from states to a probability distribution over actions, i.e. $\pi : \mathcal{S} \to P(\mathcal{A})$, which is parameterized by $\theta$. The goal of the agent is to learn a policy $\pi_\theta(a|s) = argmax_\theta[J(\pi, \mathcal{M})]$, where $J = \max_\theta \mathbb{E}_{s_0 \sim \rho(s), a \sim \pi_\theta(a_t|s_t)}[\sum_t R(s_t, a_t)]$

---

[1]For interpretabiliy, we omit orientations because center-of-mass error and orientation error were highly correlated in practice

**State Space.** The state vector $s_t = \{\phi_t, \dot{\phi}_t, \psi_t, \dot{\psi}_t, \tau_t\}$ consisted of $\phi$ a 29 dimensional vector of 23 hand and 6 arms joints and velocity $\dot{\phi}$, and object pose $\psi$ and velocity $\dot{\psi}$. In addition, positional encoding $\tau$ Vaswani et al. (2017), used to mark the current simulation timestep, was appended to the end of the state vector. This was needed for learning tasks with cyclic motions such as hammering.

**Action Space.** The action space $a_t$ was a 45-dimensional vector which consists of continuous activations for 39 muscles of wrist and fingers (to contract muscles), together with 3D translation (to allow for displacement in space), and 3D rotation of the shoulder (to allow for a wider range of arm movements).

**Reward Function.** The manipulation tasks we consider involved approaching the object and manipulating it in free air after lifting it off a horizontal surface. The hand interacts with the object adjusting its positions and orientation $X$ for a fixed time horizon. Similar to Dasari et al. (2022), this is translated into an optimization problem where we are searching for a policy that can match a desired object trajectory $\hat{X} = [\hat{x}^0, ..., \hat{x}^T]$, which is captured using the following reward function:

$$R(x_t, \hat{x}_t) := \lambda_1 exp\{-\alpha\|x_t^{(p)} - \hat{x}_t^{(p)}\|_2 - \beta|\angle x_t^{(o)} - \hat{x}_t^{(o)}|\} + \lambda_2 \mathbb{1}\{lifted\} - \lambda_3 \|\overline{m}_t\|_2 \quad (1)$$

where $\angle$ is the quaternion angle between the two orientations, $x_t^{(p)}$ is the desired object position, $x_t^{(o)}$ is the desired object orientation, $\mathbb{1}\{lifted\}$ encourages object lifting, and $\overline{m}_t$ the is overall muscle effort.

## 4.2 PREGRASP TO SIMPLIFY SEARCH SPACES

Owing to the third order non linear actuation dynamics and high dimensionality of the search space, direct optimisation of $\mathcal{M}$ leads to no meaningful behaviors. We leverage the state directly preceding the hand initiating contact with an object – i.e. pre-grasp – to greatly decrease the complexity of learning dexterous behaviors Dasari et al. (2022). Pregrasp implicitly incorporates information pertaining to the shape of the object and its associated affordance with respect to the desired task Jeannerod (1988); Santello et al. (2002). Additionally, pre-grasps can be used in our experiments without additional assumptions as they can be easily

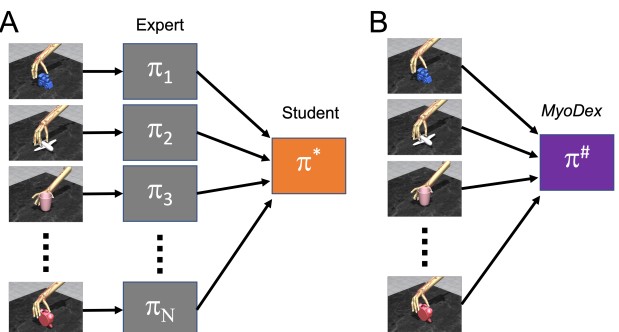

**Figure 2: Learning Paradigms.** A - Single Experts policies were obtained by training policies to solve the individual tasks. Then, by means of an expert-student approach, a unified Student policy was distilled. B - A single policy (*MyoDex*) was obtained by learning all tasks at once.

mined from MoCap recordings, annotated by human labelers, or even predicted by learned models Taheri et al. (2020a). We choose to adopt the technique from Dasari et al. (2022) and break the learning into two phases. In the first phase the hand learns to reach the pre-grasp pose using a free-space planners (no object conditioning required). Next, RL agents are trained to perform either a single target task or a family of tasks. We now describe these approaches in detail.

**Single task agents.** In our first setting, we adopt a standard RL algorithm (see 4.1) to learn a goal-conditioned policy $\pi_\theta(a_t|\phi_t, \dot{\phi}_t, \psi_t, \dot{\psi}_t, \tau_t, \hat{X}_{object}, \phi_{object}^{pregrasp})$ (see notation in Sec. 4.1) that can solve a single task. This approach will define a set of expert agents defined as $\pi_i$ with $i \in I$ where $I$ is the set of tasks (see Figure 2A).

**Multi-task agent.** Ideally, an agent would be able to solve multiple tasks using a goal conditioning variable. Thus, we additionally train a single agent to solve all 14 tasks in parallel (see Figure 2B). This approach proceeds in a similar fashion as the single-task learner, but trajectory rollouts are sampled from the 14 tasks in *parallel*. All other details of the agent $\pi_\theta^{\#}(a_t|\phi_t, \dot{\phi}_t, \psi_t, \dot{\psi}_t, \tau_t, \hat{X}_{object}, \phi_{object}^{pregrasp})$ (e.g. hyperparameters, algorithm, etc.) stay the same.

We encode manipulation behaviors in term of goal-conditioned policies $\pi_\theta(a_t|s_t)$. A standard implementation of the PPO Schulman et al. (2017) method from Stable-Baselines Raffin et al. (2021) was used. Same hyperparameters were used for all tasks (see Appendix Table A.2).

**Imitation learning.** In addition to *MyoDex* $\pi^\#$, we also train a baseline agent using $\pi^*$ expert-student method Jain et al. (2019); Chen et al. (2021) (see Figure 2A). Individual task specific policies ($\pi_i$) were used as experts. We developed a dataset with 1M samples of observation-action tuples for each of those policy. Then, we extended the observation vector to include the vector $\tau_{task}$ representing the object and trajectory. Finally, a neural network similar to Dasari et al Dasari et al. (2022) was trained via supervised learning to learn the association between observations and actions (see hyperparameters in Appendix A.1) to obtain a single policy $\pi^*(a_t|\phi_t, \dot\phi_t, \psi_t, \dot\psi_t, \tau_t, \tau_{task})$ capable of multiple task behaviors (see Figure 2A).

## 5 EXPERIMENTAL DESIGN

### 5.1 TASK DESIGN

In this study, we need a large variability of manipulations, hence it was important to include 1) objects with different shapes and weights, 2) complexity both in terms of translation and rotation of the object. Also, having different movements on the same objects allows us to investigate how the different hand pre-shapes and object trajectory affected the solution.

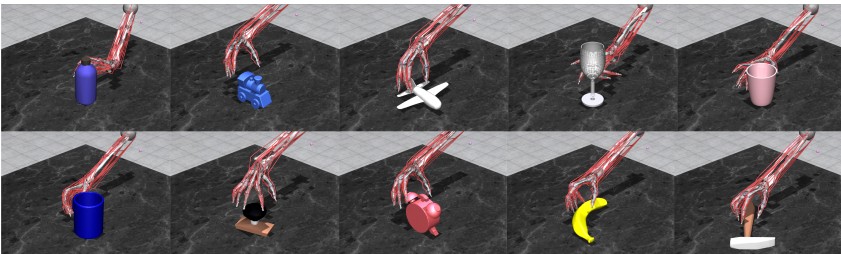

**Figure 3: A subset of *object-pregrasp* pair from our task-set.** See Table A.1 for a complete description

A set of 11 Objects and 14 different behaviors (see Table A.1) similar to the TDCM dataset presented by Dasari et al Dasari et al. (2022) were used. The setup (see Figure 3) consisted of a table-top environment, an object from the ContactDB dataset Brahmbhatt et al. (2019) and the MyoHand Caggiano et al. (2022). This dataset was implemented in the MuJoCo physics engine Todorov et al. (2012). To define our tasks, we adopted Dasari et al Dasari et al. (2022) solution where the desired object trajectory $\hat X = [\hat x^0, ..., \hat x^T]$ and the hand-object pre-grasp posture $\phi_{object}^{pregrasp} = [j_0, ..., j_n]$ where needed. We extracted these information from the GRAB motion capture Taheri et al. (2020b) dataset which contains high quality human-object interactions.

Following the same approach of Dasari et al Dasari et al. (2022), hand postures were computed by matching the human fingertip of the ContactDB dataset and MyoHand by means of Inverse Kinematics. In the context of this work, only the hand pre-shaped to grab the object before the initial contact (see Figure 3) was considered. For each task, a pre-shaped hand was used to initialize the posture of the hand and the goal was to follow a given trajectory of the object. This allow us to avoid any physical or geometric information about the object. Each tasks consisted of a pair of one trajectory of an object and its associated pre-shaped posture.

### 5.2 SYNERGY PROBING

To quantify the level of muscle coordination required for accomplishing a given task, we calculated muscle synergies by means of Non-Negative Matrix factorization (NNMF) Tresch et al. (2006). After training, we played policies for 5 roll-outs to solve specific tasks and we stored the muscle activations (value between 0 and 1) required. Then, a matrix $A$ of muscle activations over time (dimension 39 muscle x total task duration) was fed into a non-negative matrix decomposition (*sklearn*) method. The NNMF method finds two matrices $W$ and $H$ that are respectively the coefficients and the basis vectors which product approximates $A$. Muscle synergies identified by NNMF capture the

spatial regularities on the muscle activations whose linear combination minimize muscle reconstruction Bizzi & Cheung (2013). This method reveals the amount of variance explained by each of the components. We calculated the Variance Accounted For (VAF) as:

$$VAF = 100 \cdot \left(1 - \frac{(A - W \cdot H)^2}{A^2}\right) \tag{2}$$

Similarity of synergies between two different tasks was calculated using cosine similarity (CS) such as: $CS = w_i \cdot w_j$, where $[w_i, \, w_j] \in W$ are synergy coefficients respectively for the task $i$ and $j$. We used then a threshold of $0.8$ to indicate that 2 synergies were similar Appendix-A.6.

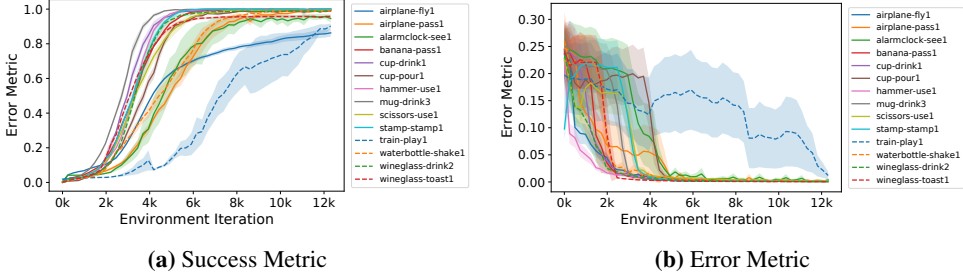

**(a)** Success Metric              **(b)** Error Metric

**Figure 4: Single task expert task-performance.** Metrics: (a) Success and (b) Error rate over iteration steps. Average and standard deviation (shaded areas) over 4 seeds are shown.

## 6 EXPERIMENTAL RESULTS

First we present how by leveraging pregrasps in standard RL pipeline, it was possible to control the physiological MyoHand to perform various tasks involving contact rich manipulation of objects (Sec.6.1). Then, we detail a deeper investigation illustrating how the muscle coordination evolves and changes as function of task conditions, learning (Sec. 6.3) and support generalization (Sec. 6.4).

### 6.1 *MyoDex*'S TASK DEXTERITY

First, we wanted to explore if we can learn a series of complex dexterous object manipulations required for performing specific tasks (see Sec. 5.1). A set of agents were trained to solve each task independently: expert solutions. The same pipeline and parameters were used to solve all tasks without any object or task-specific tuning (see Table A.2). Qualitatively, all objects in the sample were properly manipulated while moving them to follow the target trajectory (see Figure 1 for a sequence of snapshots). This was quantified by means of 2 metrics (section 3.2): Success Metric (Figure 4a) and Error Metric (Figure 4b). In all cases we achieved greater than 80% success and, overall, an error below 0.01. These analysis indicated that *MyoDex* is able to effectively drive a musculoskeletal model of the hand to learn stable object manipulations within very tight margins. To the best

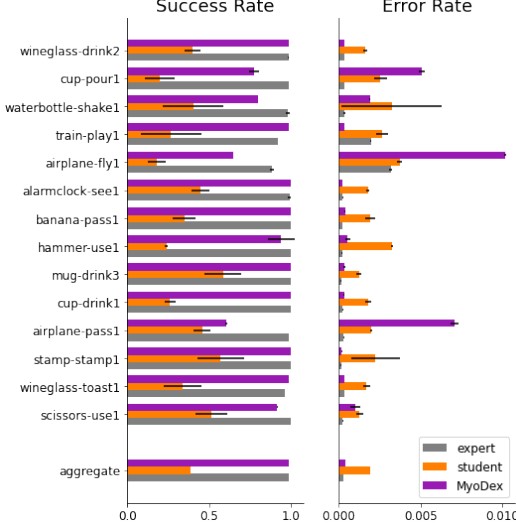

**Figure 5: Baselines**: Success and error rate metrics for Expert $\pi_i$, student $\pi^*$, and *MyoDex* $\pi^\#$ policies.

of our knowledge, this is the first demonstration of such nimble manipulation (see project website for behavior videos) with physiological musculoskeletal hand.

### 6.2 *MyoDex*'S MULTI TASK LEARNING

Next, to investigate *MyoDex* task generalization, we trained one single policy to handle all tasks simultaneously. We approached this problem in two ways. First, we used the above policies trained

to solve each single task as expert policies and, by means of an expert (teacher)-student learning paradigm, we distilled the experts into one policy that solved all tasks at the same time (see Sec.4.2). Second, by means of joint multi-task learning, we obtained one policy – *MyoDex* – that solved multiple tasks (see Sec.4.2). In both cases, the policies were able to solve the tasks but with key differences. The student policy, when compared with the *MyoDex* multi-task policy, showed a reduced success rate (see Figure 5, median success rate: expert 0.99, student 0.37, *MyoDex* 0.98) and overall greater error (see Figure 5, median error rate: expert 0.00031, student 0.00195, *MyoDex* 0.00046). In particular, the multi-task policy (see Figure A.2a) reached an error below $0.01$ in $9.1k$ iterations while the expert policies reached that error cumulatively around $34k$ iterations i.e. 4x slower. On the other hand, expert policies reached success rate of $80\%$ cumulatively in $70k$ vs $123k$ iterations needed for the *MyoDex* policy i.e. double the time. This indicates that jointly learning multiple tasks greatly facilitates the initial phase of learning, while slowing the learning of detailed aspects of each specific tasks. This is likely because tasks like *airplane-pass*, *airplane-fly* and *cup-pour* require task specific and unique wrist rotations to be accomplished.

While the student policy – obtained with imitation learning – produced muscle activations similar (Figure A.3) to that of the respective task expert (Figure A.5) but it effectiveness was quite low in task metrics.

### 6.3 DOES *MyoDex* PRODUCE REUSABLE SYNERGIES?

Biological systems simplify the problem to control the redundant and complex muscuolokeletal systems by resorting on activating particular muscle groups in consort, a phenomenon known as muscle synergies. Here, we want to analyse if synergies emerge and facilitate learning.

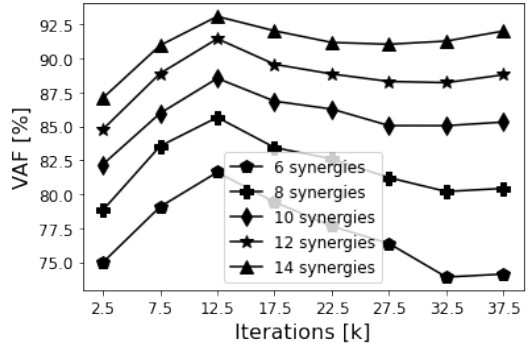

**Figure 6: Muscle Synergies over learning iterations for the joint multi-task policy.** Variance of the muscle activations (see Sec. 5.2) explained as function of the number of synergies at different steps of the learning process.

For *MyoDex* where agent has to simultaneously learn multiple manipulations / tasks, common patterns emerges and fewer synergies i.e. 12 (Figure 6), can explain the more than $80\%$ of the variance of the data (see Figure A.4). Furthermore, we observe that tasks start sharing more synergies (on average 6, see Figure A.6). This is expected as each task needs a combination of shared (task-aspecific) and task-specific synergies. Common patterns of activations seems to be related with learning. Indeed, earlier in the training more synergies are needed to explain the same amount of variance of the data. The peak is reached at $12.5k$ iterations where more than $90\%$ of the variance is explained by 12 synergies.

As expected, the expert policies shared fewer common muscle activations as indicated by fewer synergies shared between tasks (on average 2, see Figure A.6) and by the overall greater number of synergies needed to explain most of the variance: to explain more than $80\%$ of the variance it is needed to use more than 20 synergies (see Figure A.4). Similar results were obtained with the student policy (on average 1 similar synergies between tasks, see Figure A.6).

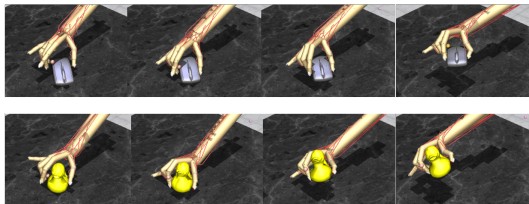

**Figure 7: Zero-shot generalization.** *MyoDex* successfully initiated manipulations on new objects and trajectories. Hand rendering includes skin (see Fig. A.1)

### 6.4 *MyoDex* OUT OF DOMAIN GENERALIZATION

The joint multi-task learning yields a policy that, by only knowing the hand posture (pre-grasp) and without information about the object, shows generalization to unseen objects and trajectories.

Qualitatively in a zero-shot out-of-domain task (Figure 8), the *MyoDex* policy initiates effective manipulations of new objects. Nevertheless, it's is not able to achieve success rates above $0.1$. The lack of complete generalization is very likely due to the missing sensory information e.g. from skin receptors, needed in order to properly hold objects with complex geometries. Indeed, in humans, when sensory information at the finger tips are inhibited, proper manipulation cannot be accomplished.

Given this initial indication of generalization capabilities, we want to explore the possibility of using the shared representation provided by the multi-task policy for 1) improving performance on the single tasks i.e. fine-tuning, 2) learning new out-of-domain tasks which were experts policies were not able to learn. We use earlier learned models i.e. $12.5k$ iterations, which provide the most general representation of coordinated movements as shown by the greater variance explained by fewer muscles synergies (see Figure 6). First, for most of the tasks, fine-tuning the multi-task representation allows faster learning of the in-domain tasks (see Table A.3). Indeed, it was possible to achieve $80\%$ success using almost less than half of the iterations ($2.8k$ vs. $5k$, fine-tuned vs experts) required for experts trained without the same model initialization. Second, we used the multi-task representation on a series of 44 new tasks (see Figure 8 and Table A.3). In most of those tasks, the shared representation allowed to quickly learn them. To be noticed, it was not possible to learn expert policies for most of those tasks without this initialization of the model. This indicates that the representation obtained by jointly learning multiple tasks helps to initialize a solution space that avoids local minima.

# 7 Conclusion

In this manuscript we showed how it is possible to control a musculoskeletal model of the human hand to learn skilled dexterous manipulation of complex objects. We were able to learn these tasks by imposing a pre-shaping of the hand that reduced the search space. In addition, by means of the joint multi-task learning we showed that it is possible to extract generalizable representations that leverage synergies – muscles that are activated as a unit – which allows both faster fine-tuning on downstream in-domain and out-of-domain tasks. All in all, this study provides strong bases for how physiologically realistic hand manipulations can be obtained by pure exploration via Reinforcement Learning i.e. without the need of motion capture data to imitate specific behaviour.

# 8 Limitations and Future work

While we have been able to show that we can produce realistic behavior without the need of fitting human data, one important limitation is understanding and matching the results with physiological data. Indeed, our exploration method via RL, produced only one of the very high dimensional combination of possible ways that a human hand could hypothetically grab and manipulate an object. For example, there are several valid ways to hold a cup e.g. by using the thumb and one or multiple fingers. Although our investigation points us in the right direction of physiological feasibility of the result, these findings have yet to be properly validated. Future works will need to consider the ability to synthesize new motor behaviors while simultaneously providing muscle validation.

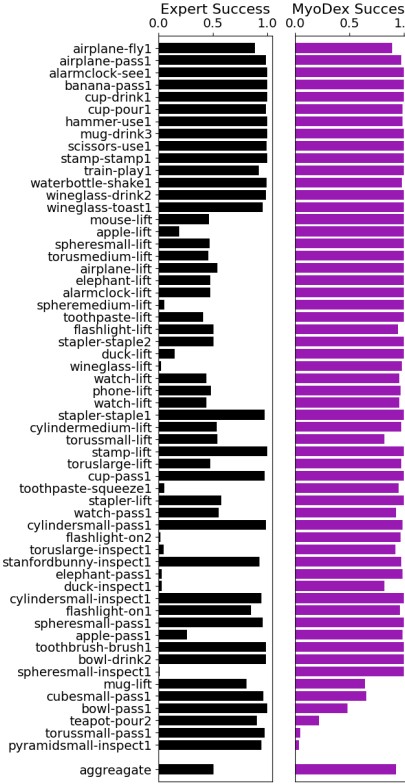

**Figure 8: Summary of all Tasks.** Left column tasks solved by single expert policies. Right columns, task fine tuning based on *MyoDex*. Aggregate success Expert vs *MyoDex* 0.51 vs 0.93. See also Table A.3.

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

# A APPENDIX

| Object | Weight [g] | behaviors |
|---|---|---|
| airplane | 172 | fly |
| airplane | 172 | pass |
| alarmclock | 542 | see |
| banana | 277 | pass |
| cup | 300 | drink |
| cup | 300 | pour |
| mug | 432 | drink |
| stamp | 210 | stamp |
| waterbottle | 364 | shake |
| wineglass | 178 | drink |
| wineglass | 178 | toast |
| train | 400 | play |
| hammer | 210 | use |
| scissors | 47 | use |

**Table A.1:** Collection of Object, weight and tasks performed on.

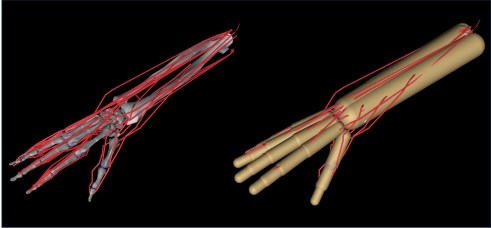

**Figure A.1:** Hand models. On the left, rendering of the musculoskeletal structure illustrating bone – in gray – and muscle – in red. On the right a skin like surfaces for soft contacts is overlaid to the musculoskeletal model.

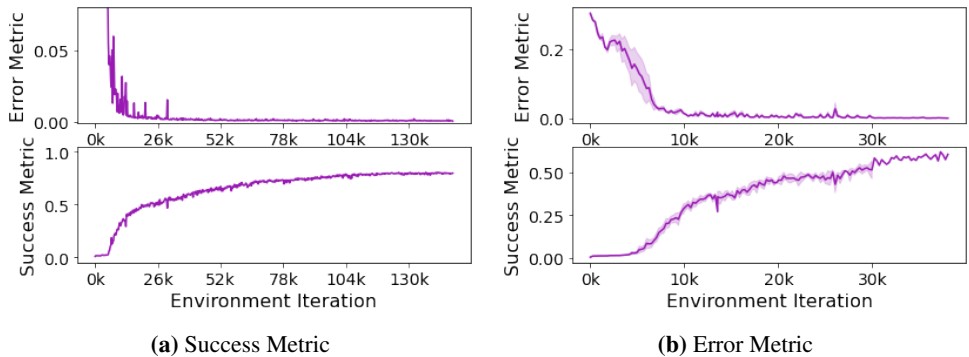

**(a)** Success Metric  **(b)** Error Metric

**Figure A.2:** Training (rollout) of the multi-task policy i.e. jointly training on all tasks. Error metrics (top) and Success metrics (bottom) illustrate that multi-task training is very efficient at reducing smaller errors. Nevertheless, overall success is achieved more slowly than expert solutions. (a) long term training (b) magnification with average - continuous line - and standard deviation - shaded area - over 5 seeds.

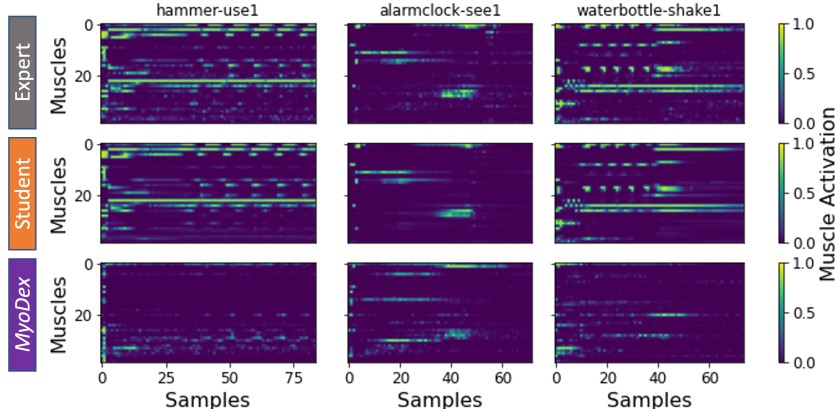

**Figure A.3: Example of muscle activations.** Expert (top) policies, student policies (middle) and multi-task/*MyoDex* (bottom).

.

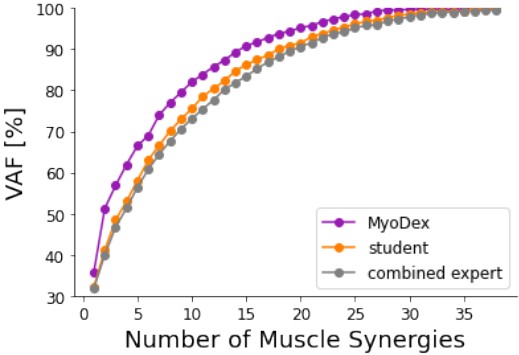

**Figure A.4: Muscle Synergies**. Number of muscle synergies as function of the explained variance (see Sec. 5.2) of the data shows that, given the same number of synergies, the multi-task learning can explain more variance of the data. Color coded experts (gray), student (orange) and *MyoDex*(purple).

A.1    PARAMETERS OF THE NEURAL NETWORK FOR THE EXPERT-STUDENT.

For distilling the single expert agents into one, a neural network of the same size of the single agent was used. We adopted a batch size 256, and Adadelta optimizer with a learning rate of $0.25$, a Discount Factor ($\gamma$) $0.995$, and 10 epochs.

| Environment Iterations | $12k$ |
|---|---|
| Discount Factor ($\gamma$) | 0.95 |
| GAE-$\lambda$ | 0.95 |
| VF Coefficient (c1) | 0.5 |
| Entropy Bonus (c2) | 0.001 |
| Clip Parameter ($\epsilon$) | 0.2 |
| Batch Size | 256 |
| Epochs | 5 |
| Network Size | $pi = [256, 128], vf = [256, 128]$ |

**Table A.2:** Parameters adopted for the reinforcement learning models.

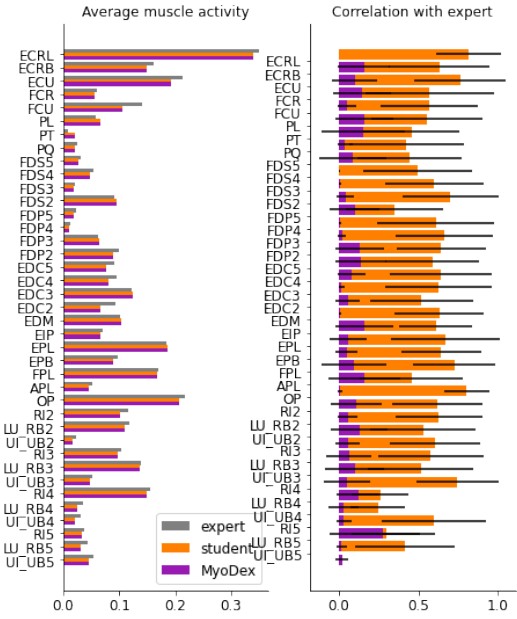

**Figure A.5:** Relationship between muscle activations. Left - Average Muscle Activation of experts (gray), student (orange) and *MyoDex*(purple). Right - correlation against the expert policies of student (orage) and *MyoDex* (purple).

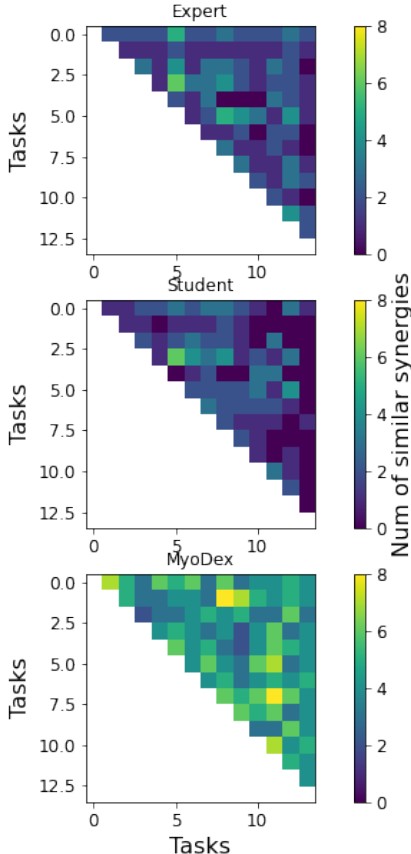

**Figure A.6:** Cosine Similarity between 12 synergies extracted from 14 different tasks at $37.5k$ iterations. Top - expert policies. Middle - student policy. Bottom – *MyoDex* policy. On average the number of similar synergies for expert, student, *MyoDex* (mean +/- std over 10 repetitions with different random seeds) was $1.88 \pm 0.9$, $1.45 \pm 0.9$ and $5.48 \pm 0.17$, respectively.

| Task | Multi-task Success | | | Iter. to reach Success of 0.8 | |
|---|---|---|---|---|---|
| | @ 1k Iter. | @ 2k Iter. | @ 3k Iter. | Multi-Task | Expert |
| stamp-stamp1 | 0.981538 | 0.997949 | 1.000000 | 247 | 3458 |
| banana-pass1 | 0.910000 | 0.993333 | 0.998571 | 247 | 4446 |
| cup-drink1 | 0.991724 | 0.998161 | 0.977471 | 247 | 3952 |
| mug-drink3 | 0.978667 | 0.999467 | 1.000000 | 247 | 3458 |
| alarmclock-see1 | 0.984444 | 0.997778 | 1.000000 | 494 | 4940 |
| train-play1 | 0.822278 | 0.929114 | 0.987848 | 741 | 8398 |
| scissors-use1 | 0.754699 | 0.945542 | 0.986988 | 1235 | 5434 |
| wineglass-drink2 | 0.714943 | 0.924138 | 0.985287 | 1235 | 4446 |
| hammer-use1 | 0.781429 | 0.870000 | 0.972857 | 1482 | 3952 |
| wineglass-toast1 | 0.713846 | 0.796410 | 0.902051 | 2223 | 4199 |
| cup-pour1 | 0.743429 | 0.730857 | 0.830286 | 2964 | 4446 |
| waterbottle-shake1 | 0.574595 | 0.709189 | 0.743784 | 3458 | 5434 |
| airplane-fly1 | 0.564675 | 0.606753 | 0.631169 | 12350 | 7657 |
| airplane-pass1 | 0.436322 | 0.497011 | 0.509425 | 12597 | 6669 |
| mouse-lift | 1.000000 | 1.000000 | 1.000000 | 247 | - |
| apple-lift | 1.000000 | 1.000000 | 1.000000 | 247 | - |
| spheresmall-lift | 0.986667 | 1.000000 | 1.000000 | 247 | - |
| torusmedium-lift | 0.980571 | 1.000000 | 1.000000 | 247 | - |
| airplane-lift | 0.995122 | 1.000000 | 1.000000 | 247 | - |
| elephant-lift | 1.000000 | 1.000000 | 1.000000 | 247 | - |
| alarmclock-lift | 1.000000 | 1.000000 | 1.000000 | 247 | - |
| spheremedium-lift | 0.998947 | 1.000000 | 1.000000 | 494 | - |
| toothpaste-lift | 0.971818 | 0.952727 | 0.990000 | 494 | - |
| flashlight-lift | 0.941714 | 0.942857 | 0.942857 | 494 | - |
| stapler-staple2 | 0.991529 | 1.000000 | 0.999529 | 494 | - |
| duck-lift | 0.994737 | 1.000000 | 1.000000 | 494 | - |
| wineglass-lift | 0.933000 | 0.979500 | 0.980000 | 494 | - |
| watch-lift | 0.925333 | 0.955556 | 0.955556 | 741 | - |
| phone-lift | 0.960000 | 0.967742 | 0.967742 | 741 | - |
| watch-lift | 0.825778 | 0.953778 | 0.955556 | 988 | - |
| stapler-staple1 | 0.893809 | 0.989524 | 0.996190 | 988 | 5187 |
| cylindermedium-lift | 0.841111 | 0.970000 | 0.972222 | 988 | - |
| torussmall-lift | 0.690285 | 0.931428 | 0.915428 | 1235 | - |
| stamp-lift | 0.709756 | 0.980488 | 0.992195 | 1235 | 3211 |
| toruslarge-lift | 0.707273 | 0.965455 | 0.977273 | 1235 | - |
| cup-pass1 | 0.609048 | 0.995238 | 1.000000 | 1235 | 4446 |
| toothpaste-squeeze1 | 0.598421 | 0.943157 | 0.977368 | 1482 | - |
| stapler-lift | 0.650732 | 0.868293 | 0.982439 | 1482 | - |
| watch-pass1 | 0.492593 | 0.887407 | 0.884444 | 1729 | - |
| cylindersmall-pass1 | 0.571200 | 0.826667 | 0.901333 | 1976 | 4693 |
| flashlight-on2 | 0.168791 | 0.695385 | 0.920000 | 2470 | - |
| toruslarge-inspect1 | 0.251852 | 0.645926 | 0.817778 | 2470 | - |
| stanfordbunny-inspect1 | 0.289157 | 0.591325 | 0.921446 | 2470 | 6422 |
| elephant-pass1 | 0.506667 | 0.621235 | 0.834568 | 2964 | - |
| duck-inspect1 | 0.621299 | 0.624935 | 0.820260 | 2964 | - |
| cylindersmall-inspect1 | 0.420000 | 0.713333 | 0.639444 | 3705 | 6422 |
| flashlight-on1 | 0.234483 | 0.541609 | 0.626207 | 4446 | 10127 |
| spheresmall-pass1 | 0.191905 | 0.351905 | 0.674286 | 4446 | 5928 |
| apple-pass1 | 0.344198 | 0.481481 | 0.583210 | 5187 | - |
| toothbrush-brush1 | 0.119375 | 0.353125 | 0.589063 | 5434 | 4199 |
| bowl-drink2 | 0.075714 | 0.089524 | 0.163810 | 7657 | 4693 |
| spheresmall-inspect1 | 0.235676 | 0.332432 | 0.438919 | 8151 | - |
| mug-lift | 0.326575 | 0.335342 | 0.397808 | - | 7904 |
| cubesmall-pass1 | 0.024691 | 0.024691 | 0.024691 | - | 5928 |
| bowl-pass1 | 0.114430 | 0.153418 | 0.184810 | - | 7163 |
| teapot-pour2 | 0.137627 | 0.150508 | 0.162712 | - | 7657 |
| torussmall-pass1 | 0.038987 | 0.037975 | 0.037975 | - | 5928 |
| pyramidsmall-inspect1 | 0.028571 | 0.033333 | 0.035238 | - | 5187 |

**Table A.3: Fine-tuning of 58 different tasks for *MyoDex* and expert agents.** Expert solutions could reliably reach 0.80 success for the first 14 tasks but in many other cases they were not able to. A few exceptions at the bottom show success only for expert solutions. We indicated with '-' the lack of success in achieving the success threshold. The first 3 columns report the success rate respectively at 1k, 2k and 3k iterations. The 4th and 5th column, document the iterations at which 0.80 success for *MyoDex* and experts was reached.

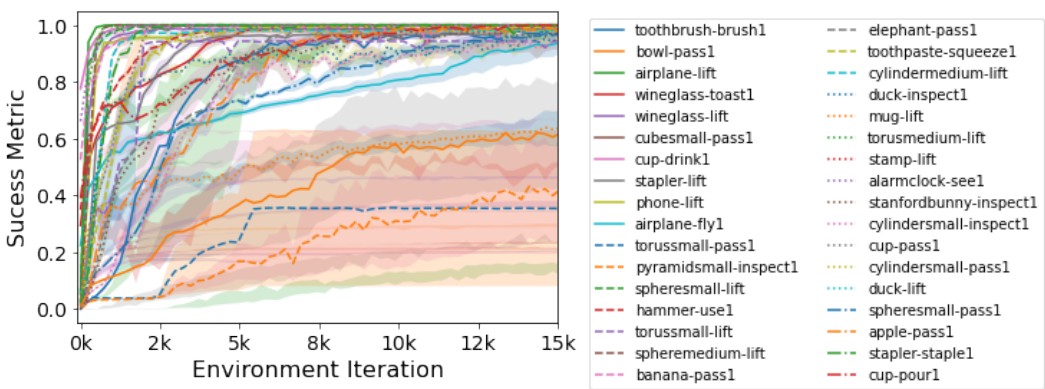

**Figure A.7: Finetuning *MyoDex* on a large set of tasks.** Success rate over iterations of finetuning *MyoDex*. Shaded areas indicate standard deviation over 3 seeds.

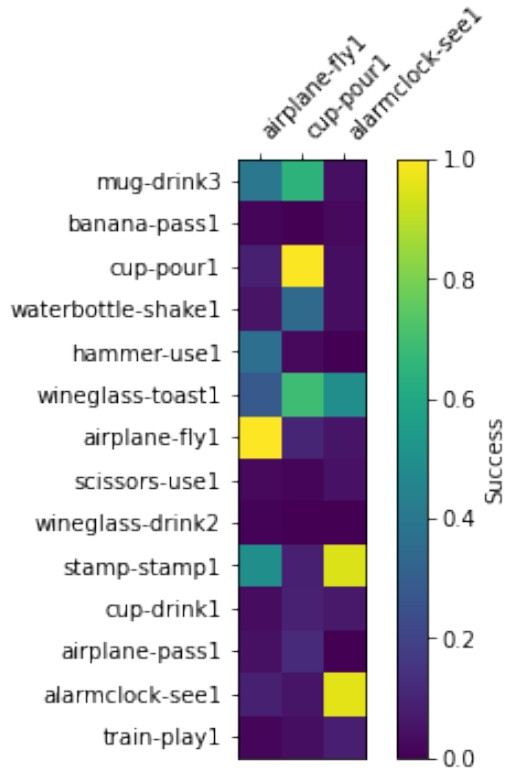

**Figure A.8: Fine-tuning based on expert policies.** Success rate fine-tuning experts solutions (columns) on 14 different environments. This matrix shows that the combination of pre-grasps and the initialization on a pre-trained task is not enough to generalize to new tasks.

