# OpenReview forum: "MyoDex: Generalizable Representations for Dexterous Physiological Manipulation"
_ICLR.cc/2023/Conference — Submitted to ICLR 2023_

### Official Review · Reviewer_G4dn · 2022-10-22

**Confidence:** 3
**Correctness:** 3
**Technical Novelty And Significance:** 2
**Empirical Novelty And Significance:** 2
**Recommendation:** 6

**Clarity, Quality, Novelty And Reproducibility:**

The method seems straightforward and clearly described.

To truly facilitate reproducibility, it would be best if the authors can release the code, since there are also some complication coming from defining the pregrasp pose and desired trajectories for the objects.

**Strength And Weaknesses:**

Pro:

1. A simple system that can learn a wide range of manipulation tasks for a musculoskeletal hand. This can serve as baseline for future research in this direction.

2. Demonstration of the advantage of learning multiple tasks at the same time, using metrics like synergies and task generalization.

Con:

1. There are some problems about the comparison in Table I, see comment below.

2. The failure of the student policy to learn effectively from expert policies need more analysis.

**Summary Of The Paper:**

This paper presents s framework to learn manipulation skills with a musculoskeletal hand. Furthermore, this paper demonstrates that a multi-task policy can provide better generalization, e.g., better measure in term of synergies and better generalization to new tasks.

**Summary Of The Review:**

This paper provides a good baseline for learning musculoskeletal hand control. The findings that multitask learning help better generalization is interesting. Some questions to addressed during the rebuttal:

1. The comparison in Table I seems unfair, since MyoDex is initialized with some pretrained policy and the expert is trained from scratch, isn't it expected that MyoDex will win? And it is interesting that in the final two cases of the in distribution task, the expert actually wins. A more fair comparison might be to also taking into account the number of iterations used during MyoDex initial training.

2. It is unclear why the expert fails all the out-of-domain tasks. Is it because these tasks are fundamentally harder? How are the initial set of task to train the expert/MyoDex selected? It will be nice to provide some intuition. Does decreasing/increasing the set of initial tasks affect the performance of the multitask learned policy? i.e., how do we know what tasks to train for initially?

3. Why does the student policies fail to match the performance of the expert policies? Have the authors tried DAGGER? It is unclear to me whether it is due to fundamental challenge to imitate a large number of experts or if it is a purely implementation issue. More evidence is needed to support the claim. e.g., Are the student able to just imitate one expert effectively? What about two? What is the minimum number of experts that will cause the student to fail to learn effectively?

---

> ### Author Response · Authors · 2022-11-11
> **Point to point (1/2)**
>
> We thank the reviewer for their constructive comments and positive assessment of our work! Here, we respond to the questions and comments the reviewer raises. Please let us know if we can provide any additional clarifications during the discussion period to improve our score.
>
> > This paper provides a good baseline for learning musculoskeletal hand control. The findings that multitask learning help better generalization is interesting. Some questions to addressed during the rebuttal:
> > Q1. The comparison in Table I seems unfair, since MyoDex is initialized with some pretrained policy and the expert is trained from scratch, isn't it expected that MyoDex will win? And it is interesting that in the final two cases of the in distribution task, the expert actually wins. A more fair comparison might be to also taking into account the number of iterations used during MyoDex initial training.
>
> R1: We apologize for creating this confusion. The goal is not to show that MyoDex is the winning solution against experts or student policies. Rather to show that learning multiple tasks can deliver general behavioral priors. When encountered with new unseen tasks, if the priors are indeed general, we would observe accelerated learning, and inductive biases that avoid local minima. Indeed, we observe better sample complexity (compared to learning from scratch) for new tasks; even solve tasks that otherwise can’t be solved from scratch.
>
> Actually, it is not obvious at all that learning together tasks will result in greater generalization capabilities. One obvious expectation was that learning some tasks brings the network toward a solution which is incompatible in supporting unseen tasks whose solution might lie in a completely different state manifold. Indeed, we observe this in a few tasks (bottom of Figure 8) where we can find expert solutions but no MyoDex solutions.
>
> We want to clarify also that finetuning started with MyoDex solutions at 12.5k iterations (where we found the higher generalization of synergies, see Figure 6) and policies fine tuned based on MyoDex were much faster at reaching high success rate: it was possible to achieve 80% success using almost less than half of the iterations (2.8k vs. 5k, fine-tuned vs experts, see Table A.3 comparing the last 2 columns).
>
> > Q2. It is unclear why the expert fails all the out-of-domain tasks. Is it because these tasks are fundamentally harder?
>
> R2: Our tasks are sampled from a large heterogeneous set of daily activities that constitutes a combination of in-hand and full arm manipulation and are critically important for tasks of daily living. Given the task diversity and dimensionality of the actuated musculoskeletal systems, individual experts' solutions lie in narrow manifolds that are unlikely to generalize to new out of domain tasks (see Figure A.8). In contrast, MyoDex representations generalize to unseen tasks. We have added 30 more tasks (for a total of 58) in the revised version of the manuscript to remark this point.
>
>
> > Q3. How are the initial set of tasks to train the expert/MyoDex selected? It will be nice to provide some intuition.
>
> R3: We empirically tested a few combinations of multi-task (4, 6) policies. We then chose the tasks for which we found consistent expert solutions across various sets. Our final set consists of 14 different tasks. The set of 14 strikes the balance between multi-task complexity, computation, and resulting generalization.
>
>
> > Q4: Does decreasing/increasing the set of initial tasks affect the performance of the multitask learned policy? i.e., how do we know what tasks to train for initially?
>
> R4: As mentioned in the previous point, we started with a few combinations. Nevertheless, this is a great point and will require an in-depth study. At this point, because of the lack of a comprehensive metric that allows us to establish the task complexity with respect to the pregrasp, it is impossible to define a range of complexity on the definition of MyoDex. The only alternative solution at this moment is to consider all the possible factorial combinations of tasks. Besides the computational complexity, the solution to this process will be very difficult to interpret because of the lack of the above complexity metrics. We have added 30 more tasks (for a total of 58) in the revised version of the manuscript to further analyze the generalization capability of MyoDex.

---

> > ### Author Response · Authors · 2022-11-11
> > **Point to point (2/2)**
> >
> > > Q5. Why does the student policies fail to match the performance of the expert policies? Have the authors tried DAGGER? It is unclear to me whether it is due to fundamental challenge to imitate a large number of experts or if it is a purely implementation issue. More evidence is needed to support the claim. e.g., Are the student able to just imitate one expert effectively? What about two? What is the minimum number of experts that will cause the student to fail to learn effectively?
> >
> > R5: The student policy is able to capture overall behaviors of the experts (see both muscle activation Figure A.3 and Errors Figure 5), but fail to capture the specific nuances. This is understandable since all experts are learned separately without any coordination or consistency between them in terms of behavioral priors. They learn very specific solutions for the tasks and do not share coordinated patterns among them (this can be seen in the shared synergies of experts in Figure A.6). Reviewer is correct that the discrepancy arises from trying to capture disjoint modes of individual experts into a single network. On the other hand MT policies reinforce priors that are shared across tasks. Solving multiple tasks at once forces coordination resulting in behavioral priors that are implicitly captured during training (Figure 6 and A.2).
> >
> >
> > Please let us know if our response answers the reviewer’s questions. We are happy to provide additional clarifications to improve our score. We thank the reviewer again for their time and effort helping us improve our paper!

---

### Official Review · Reviewer_Sq1z · 2022-10-23

**Confidence:** 4
**Correctness:** 3
**Technical Novelty And Significance:** 2
**Empirical Novelty And Significance:** 3
**Recommendation:** 5

**Clarity, Quality, Novelty And Reproducibility:**

* The methodological contributions are limited, borrowing heavily from the Desari et al., 2022. However, this could be made more clear.
* Extension of the MyoHand environment to include the shoulder is a novel contribution
* The authors claim that 14 of the tasks in the paper were previously unsolved. Hence, the application of the method to these tasks represents empirical novelty and significance.
* The paper is at time unclear with sections that logically may go together being separate (e.g., 4.2 and 5.2). Additionally several experimental details and protocols are unclear.
* Only one seed seems to be used in evaluation calling into question the reproducibility and stability of the results.

**Strength And Weaknesses:**

**Strengths**

*S1.* The problem of controlling small, dexterous muscle groups seems understudied and this paper takes a step in understanding how techniques might be applied to address this problem.

*S2.* Experiments are creative and well intentioned. For example, probing for synergies between muscle groups is an interesting idea, with possible implications for transfer-learning on hand models.

*S3.* Extension of the MyoHand simulation to include shoulder movement.

*S4.* Empirical novelty demonstrating the application of Pre-grasp + PPO to the manipulation tasks considered in the evaluated benchmark.

**Weaknesses**

*W1.* The task formulation is not entirely clear (Sec. 3.2). Currently it just presents metrics, which is different than the task itself. Also it is not clear why there are both "Task Formulation" and "Problem Formulation" (Sec. 4.1) sections.

*W2.* What is $\phi$ in Sec. 4.1? What is the dimensionality? How does it relate to the hand model? What is meant by a "joint angle" in this case?

*W3.* The action space is not clear in the manuscript. What is an example of an action? What does an action represent? Some continuous contractions of muscles? What is an example of a state?

*W4.* The reward function used this this work looks to be the same as that in Dasari et al., 2022. The authors must cite this to make this attribution clear.

*W5.* The method assumes a lot of information about the scene (1) GT state (2) densely annotated ground truth trajectories for their dense reward (3) does not consider complications associated with vision. I consider these to be limitations of the work in its current state, but still feel the exploration is valid.

*W6.* The training dataset details are not clear in the manuscript. For example for each task how many GT training trajectories are sampled? A single trajectory? Multiple trajectories? Given a single task, are the target objects always spawned in the same pose? The manuscript discusses pose variations between different tasks but it is unclear if there is any randomization when training one task.

*W7.* The pre-grasp, introduced in Dasari et al., 2022. seems to greatly simplify the problem and search space. However, I consider pre-grasp as privileged information and wonder if the pre-grasp makes grasping the object trivial. Given the pre-grasp what is the challenge of closing the MyoHand?

*W8.* Only one seed seems to be used in evaluation calling into question the reproducibility and stability of the results.

*W9.* In the explanation of VAF metric (Sec. 5.3) it is not clear on what matrix NNMF is done on. What is the matrix A exactly and what is its dimensions? Is it the output of your network? Perhaps including an example here would add clarity.

*W10.* In Sec. 5.3, the use of cosine similarity to count the number of synergies is not completely convincing to me. My understanding is that H is a matrix of basis vectors and and W of coefficients for the basis vectors for a task. Now consider a permutation of the basis H, which leads to a permutation of W. This may change the cosine similarity when comparing two synergies.

*W11.* It appears that $\pi^*$ does not have access to the pre-grasp, while MyoDex does. I consider a policy "$\pi^*$ + pre-grasp" to be a more fair point of comparison as it has access to the same amount of privileged information.

*W12.* No network architecture details are presented. It is unclear how big (or small) the networks are. I suggest adding more details about the network to the main paper or appx.

*W13.* It is hard to contextualize what a synergy is in Fig. 6 and what exactly 12 synergies means. Does this map to muscles firing together? Are there interpretable examples of these synergies that can be observed in the resulting behavior? Is it possible to have a human or scripted policy baseline here to contextualize what 12 synergies means?

*W14.* The zero-shot protocol is not clear. MyoDex is conditioned on the goal object. What is the protocol for conditioning on an unseen object? How is the pre-grasp provided? I recommend adding details to make this section more clear.

*W15.* How were multi-task and transfer task splits chosen? Are the tasks taken from the TCDM benchmark introduced in Dasari et al., 2022? If so, I strongly recommend citing the benchmark and making the benchmark setting details more transparent.

*W16.* There are many holes in Tab. 1. Is it the case that these policies did not reach 0.8 success for these tasks? If so, I recommend making this more clear in the caption, so it is clear that these evaluation were actually run. Also I recommend reporting the success rate curves in the appx. so that future work can try to improve on this task subset.

**Minor**

*M1.* Consider not splitting 4.2 and 5.2 into different sections

*M2.* 6.2 is not necessarily about task generalization, but about multi-task experiments. I feel these are different things and the title should be adjusted accordingly.


**Summary Of The Paper:**

The paper presents the MyoDex framework, which applies goal conditioned pre-grasping and PPO (introduced in prior work) to grasping and manipulation tasks with a simulated human hand. The paper considers multi-task and transfer-learning scenarios showing the superiority of the method over baselines.

**Summary Of The Review:**

The paper explores the application of a method to grasp and manipulate objects with a biologically accurate hand model. The authors provide empirical evidence on the effectiveness of this method and additionally aim to understand the use of the method for multi-task-learning, transfer-learning, and qualitatively probe the resulting behaviors for synergies among muscle groups.

However, I currently recommend weak rejection of the manuscript. I am most concerned about the clarity of the manuscript on key methodological and experimental details (*W1-3, 6, 9, 10, 12-15*), the fairness of the multi-task baseline (*W11*), and the lack of attribution to the Dasari et al., 2022 paper on key aspects of the methodological and task (*W4, 7, 15*). Additionally, I am not convinced that the task is non-trivial (*W7*).

I am willing to revisit my evaluation during the rebuttal/discussion period.

---

> ### Author Response · Authors · 2022-11-10
> **Point to point (1/3)**
>
> We thank the reviewer for the constructive comments and the overall positive assessment of our contributions. We have addressed the reviewer specific comments below. Please let us know if we can provide any additional clarifications during the discussion period to improve our score.
>
> > W1. The task formulation is not entirely clear (Sec. 3.2). Currently it just presents metrics, which is different than the task itself. Also it is not clear why there are both "Task Formulation" and "Problem Formulation" (Sec. 4.1) sections.
>
> R1: Please note the section is titled “Task Formation” (not formulation). This section outlines the task (and evaluation metrics) independent of the solution strategy. Dexterous manipulation is often posed as a problem of achieving the final configuration of the object. In this study we are interested to capture the whole continuous aspects of the manipulation behaviour with object maneuver e.g., drinking, playing, even cyclic  like hammering. Those tasks are hard to capture as goal reaching. To effectively capture the temporal behaviors, we instead define dexterous manipulation as a task of realizing a desired object temporal trajectory ($\hat{X}$).
>
> In the “Problem Formulation” section we outline how MyoDex approaches the tasks inside a MDP formulation. Clean separation between the task definition and how we approach it removes any algorithmic biases and ensures generality of our task definition and task suite.
> We renamed the section to “Task” and added clarifications to avoid this confusion.
>
> > W2. What is ϕ in Sec. 4.1? What is the dimensionality? How does it relate to the hand model? What is meant by a "joint angle" in this case?
>
> R2: ϕ refers to the joint angles of the hand and arms in radians. We use joint definitions as outlined in MyoSuite [Section 3.1 Caggiano et al 2022] models. MyoSuite models are validated for physiological accuracy both in terms of kinematics details of the joints and as well as actuation details of the muscle dynamics. In our tasks ϕ is 29 dimensional and constitutes of 23 hand and 6 arms joints. We added these details in Section 4.1
>
> > W3. The action space is not clear in the manuscript. What is an example of an action? What does an action represent? Some continuous contractions of muscles? What is an example of a state?
>
> R3: The action space $a_t$ is a 45-dimensional vector which consists of continuous activations for 39 muscles of wrist and fingers (to contract muscles), together with 3D translation (to allow for displacement in space), and 3D rotation of the shoulder (to allow for a wider range of arm movements).
>
>
> > W4. The reward function used this this work looks to be the same as that in Dasari et al., 2022. The authors must cite this to make this attribution clear.
>
> R4: We leveraged the same approach as for Dasari et al. both for the tasks and reward functions. The work is cited in Sections 4.1 and 4.2. We have added additional citations to this reference in Section 5.1.
>
>
> > W5. The method assumes a lot of information about the scene (1) GT state (2) densely annotated ground truth trajectories for their dense reward (3) does not consider complications associated with vision. I consider these to be limitations of the work in its current state, but still feel the exploration is valid.
>
> R5: The primary objective of our work is to retrieve behavioral priors that would allow generalization. The only information used were the trajectory of the object and the hand pre-shape. We focus on challenges presented by high dimensionality of action space. To avoid confounding challenges from representation learning in high dimensional input space (vision, tactile), we present and analyze our findings in low level state space.
>
>
> > W6. The training dataset details are not clear in the manuscript. For example for each task how many GT training trajectories are sampled? A single trajectory? Multiple trajectories? Given a single task, are the target objects always spawned in the same pose? The manuscript discusses pose variations between different tasks but it is unclear if there is any randomization when training one task.
>
> R6: We apologize for this confusion. We used only a single object trajectory associated with each pregrasp. We better clarify this point in the manuscript in Section 5.1

---

> > ### Author Response · Authors · 2022-11-10
> > **Point to point (2/3)**
> >
> >
> > > W7. The pre-grasp, introduced in Dasari et al., 2022. seems to greatly simplify the problem and search space. However, I consider pre-grasp as privileged information and wonder if the pre-grasp makes grasping the object trivial. Given the pre-grasp what is the challenge of closing the MyoHand?
> >
> > R7: Only access to pregrasp does not lead to success in sample inefficient samples. Indeed, we showed that individually trained policies are not able to learn complex tasks just based on this a priori information [28 unsolved tasks with experts, Figure 8 and Table A.3]. Also, closing the hand is not enough to solve the task. Our tasks require a complex coordination of heterogeneous (redundant and multiarticular) muscles for in-hand reorientation as well as full arm motion behaviors. If “grasp and move”  was the enough to solve the task, it would have trivially emerged from single task expert policy. Nevertheless, single expert policies were unsuccessful in 28 tasks. Pre-grasp was shown that the method can work with predicted gasps [see Table 1 of Dasari et al 2022]. In this work we assumed pre-grasp as given to focus on our contributions. We however would emphasize that this experimentation choice can be lifted using predicted grasp.
> >
> > > W8. Only one seed seems to be used in evaluation calling into question the reproducibility and stability of the results.
> >
> > R8: We have evaluated multiple seeds (see Figure 4, A.2, A.7). We see very low seed to seed variance which indicates the stability of behavior induced by MyoDex.
> >
> > > W9. In the explanation of VAF metric (Sec. 5.3) it is not clear on what matrix NNMF is done on. What is the matrix A exactly and what is its dimensions? Is it the output of your network? Perhaps including an example here would add clarity.
> >
> > R9: We better explained how we calculated muscle synergies in section 5.2 “We collected 5 trajectories per task using either expert, student, or MyoDex policies. The muscle activations signals from these trajectories were used to construct the matrix $A$ which consists of muscle activations over time (dimension 39 muscles x total task duration). This was fed into a non-negative matrix decomposition (\textit{sklearn}) method. The NNMF method finds two matrices $W$ and $H$ that are respectively the coefficients and the basis vectors whose product approximates $A$.”
> >
> > > W10. In Sec. 5.3, the use of cosine similarity to count the number of synergies is not completely convincing to me. My understanding is that H is a matrix of basis vectors and and W of coefficients for the basis vectors for a task. Now consider a permutation of the basis H, which leads to a permutation of W. This may change the cosine similarity when comparing two synergies.
> >
> > R10: The reviewer is right that the permutation can affect the cosine similarities. We have added permutations on 10 random seeds (Figure A.6) which show that the similarity is maintained although independently from the initialization and independently from the permutation: mean +/- std  was $1.88\pm0$.9, $1.45\pm0.9$ and $5.48\pm0.17$ for expert, student, and MyoDex, respectively (see also Figure A.6 where we compare all the possible combinations of pairs of muscles).
> >
> > > W11. It appears that π∗ does not have access to the pre-grasp, while MyoDex does. I consider a policy "π∗ + pre-grasp" to be a more fair point of comparison as it has access to the same amount of privileged information.
> >
> > R11: We would like to clarify that this is incorrect. The experts that are used to gather the data were provided access to and exploited the pre-grasps as well. π∗ is only distilled from experts but it has access to the pregrasp as initialization of the environment during testing.
> >
> > > W12. No network architecture details are presented. It is unclear how big (or small) the networks are. I suggest adding more details about the network to the main paper or appx.
> >
> > R12: We have used similar architecture as for Dasai et al 2022. We referred to the network architecture in the appendix (Table A2).
> >
> > > W13. It is hard to contextualize what a synergy is in Fig. 6 and what exactly 12 synergies means. Does this map to muscles firing together? Are there interpretable examples of these synergies that can be observed in the resulting behavior? Is it possible to have a human or scripted policy baseline here to contextualize what 12 synergies means?
> >
> > R13: Yes, the reviewer is right: muscle synergies are weighted co-activations of muscles. It is challenging to see the synergies in action but to show the overall behavior we have added a movie on the project’s page https://sites.google.com/view/myodex/. In this movie, a sequence of 12 synergies were played by considering a continuous maximum activation of all coefficients.

---

> > > ### Author Response · Authors · 2022-11-11
> > > **Point to point (3/3)**
> > >
> > >
> > > > W14. The zero-shot protocol is not clear. MyoDex is conditioned on the goal object. What is the protocol for conditioning on an unseen object? How is the pre-grasp provided? I recommend adding details to make this section more clear.
> > >
> > > R14: While testing on a new task/object, we provide only the desired object trajectory for the object and the hand pregrasp pose. Note that we do not use any information about the object.  We have clarified this point in the manuscript in Section 5.1.
> > >
> > > > W15. How were multi-task and transfer task splits chosen? Are the tasks taken from the TCDM benchmark introduced in Dasari et al., 2022? If so, I strongly recommend citing the benchmark and making the benchmark setting details more transparent.
> > >
> > > R15: We chose 14 of the most articulated tasks where we found consistent expert solutions as the reference for the multi-task learning of MyoDex. Tasks were taken from the same dataset used to build the Dasari et al 2022 manuscript. We clarified this point in Section 5.1 of the manuscript and added proper citations.
> > >
> > > > W16. There are many holes in Tab. 1. Is it the case that these policies did not reach 0.8 success for these tasks? If so, I recommend making this more clear in the caption, so it is clear that these evaluation were actually run. Also I recommend reporting the success rate curves in the appx. so that future work can try to improve on this task subset.
> > >
> > > R16: We added the success rate curves in the appendix, we moved the table (extended with more tasks) into the appendix with an extended legend. Also, we added the success rate curves (Figure A.7)
> > >
> > > > M1. Consider not splitting 4.2 and 5.2 into different sections
> > >
> > > RM1: we merged sections 4.2 and 5.2
> > >
> > > > M2. 6.2 is not necessarily about task generalization, but about multi-task experiments. I feel these are different things and the title should be adjusted accordingly.
> > >
> > > RM2: We thank the reviewer for this suggestion. We have adjusted the title for clarity.
> > >
> > > Please let us know if our response answers the reviewer’s questions. We are happy to provide additional clarifications to improve our score. We thank the reviewer again for their time and effort helping us improve our paper!

---

> > > > ### Comment · Reviewer_Sq1z · 2022-11-20
> > > > **Post rebuttal**
> > > >
> > > > Thank you to the authors for their responses and additional analysis. However, I feel comfortable with my original score for the following reasons:
> > > > 1. from reading the revised manuscript key details are still unclear. For example, it is not clear that the method "provide[s] ... the desired object trajectory," which was only made clear to me in the rebuttal text. However, this raise the question--if the GT state of the target object is known, and the trajectory is taken is input, why is this a learning problem? Can't a solution be found using a planning approach?
> > > >
> > > > 2. I still do not have a great understanding of why expert policies should fail.
> > > >
> > > > 3. One of the key contributions is related to discovering synergies; however, it is still not clear to me what a synergy is.
> > > >
> > > > 4. In the rebuttal the authors emphasize "generalization"; however, I do not consider the fine-tuning experiments to be indicative of "generalization" capabilities.

---

### Official Review · Reviewer_sdCp · 2022-10-24

**Confidence:** 4
**Clarity, Quality, Novelty And Reproducibility:** The paper is well-written and all the…
**Correctness:** 3
**Technical Novelty And Significance:** 2
**Empirical Novelty And Significance:** 2
**Recommendation:** 5

**Strength And Weaknesses:**


Strength:
1.	The proposed contact-rich manipulation tasks for the musculoskeletal dexterous agent are novel.
2.	MyoDex leverages joint multi-task learning to recover reusable representations (synergies) that allow for easier fine-tuning in both in-domain and out-of-domain tasks (including one/few-shot learning).

Weaknesses
1.	All the tasks in this paper seem based on grasping. The difficulties of these tasks are too homogeneous. Perhaps refer to these projects to design more tasks, such as [1] and [2].
2.	The State Space contains the pose and velocity of the object without any vision-based input, thus resulting in missing geometric features of the object. This may limit the out-of-domain generalization ability. Recently, lots of dexterous hand works contain visual input with a strong generalization ability, such as [3] and [4].
3.	There seem to be no ablation experiments on synergies.
4.	The experiment seems to be done with only one seed, and the authors should consider doing several more seeds in order to better demonstrate the robustness of the method.
5.	The previous work [5], which also presents a dexterous manipulation benchmark with musculoskeletal hands, detracts from the novelty of this work.

[1] https://bi-dexhands.ai/
[2] https://openreview.net/pdf?id=k2Ml8FGtJZp
[3] https://openreview.net/pdf?id=k2Ml8FGtJZp
[4] https://openreview.net/pdf?id=tJE1Yyi8fUX
[5] https://arxiv.org/pdf/2205.13600.pdf


**Summary Of The Paper:**

This paper presents a benchmark that touches on an interesting and important topic, musculoskeletal dexterous hand manipulation. The authors further demonstrate that pre-training the policy on 14 manipulation tasks allows for easier fine-tuning in both in-domain and out-of-domain tasks.

**Summary Of The Review:**

Overall, the paper is of good quality. However, the main contribution of this work is to demonstrate the effectiveness of pretraining the musculoskeletal dexterous agent, which I think is not enough for acceptance. I would consider increasing my rating if the authors could address my concerns.

---

> ### Author Response · Authors · 2022-11-10
> **Point to point response**
>
> We thank the reviewer for finding the dataset, tasks and approach novel. We have addressed below the questions raised in a point-to-point response. Please let us know if we can provide any additional clarifications during the discussion period to improve our score.
>
> > Q1. All the tasks in this paper seem based on grasping. The difficulties of these tasks are too homogeneous. Perhaps refer to these projects to design more tasks, such as [1] and [2].
>
> R1: We respectfully disagree with the reviewer that the current tasks are too homogenous. Our tasks are sampled from a large heterogeneous set of daily activities that constitutes a combination of in-hand and full arm manipulation and are critically important for tasks of daily living. Additionally, the degrees of complexity to hold the control object to achieve complex rotations are very challenging for high dimensional over actuated musculoskeletal systems. Indeed, it requires a wide range of stability in coordinating muscle activations for manipulating the object and performing the wrist rotation. For this reason, SOTA results are typically restricted to basic reaching/ grasping. This is the first time that contact rich hand-object interactions of such high complexity have been demonstrated with musculoskeletal systems.
>
> In the revised manuscript we have further boosted the task complexity by adding 3x more tasks further demonstrating MyoDex’s generalization abilities.
>
> > Q2. The State Space contains the pose and velocity of the object without any vision-based input, thus resulting in missing geometric features of the object. This may limit the out-of-domain generalization ability. Recently, lots of dexterous hand works contain visual input with a strong generalization ability, such as [3] and [4].
>
> R2: We share the reviewer point and we agree that generalization can be further improved by tactile and/or visual information. The primary objective of our work is to retrieve behavioral priors that would allow generalization. In this work, we focus on challenges presented by the high dimensionality of “action space”. To avoid confounding challenges from representation learning in high dimensional “input-space” (vision, tactile), we present and analyze our findings in low level state space. Building up from here and introducing representational challenges from high dimensional input spaces is our future work.
>
> > Q3. There seem to be no ablation experiments on synergies.
>
> R3: In this study we do not constrain control based on synergies. We observe that synergies implicitly emerge during jointly learning multiple tasks. One possible way to see the effect of ablating/removing synergies would be to fine tune from individual experts that have a limited number of synergies e.g. expert solutions. Indeed, using information from single policies (experts) to fine tune additional tasks (see Figure A.8) doesn’t lead to the same generalization as observed with MyoDex. Out of 14 tasks, only 2 tasks benefitted from fine-tuning the experts vs all of them by fine-tuning MyoDex (see Figure A.8 and Figure 8).
>
> > Q4. The experiment seems to be done with only one seed, and the authors should consider doing several more seeds in order to better demonstrate the robustness of the method.
>
> R4: We added 3 more seeds to the results (see Figure 4, A.2, A.7) which show little variability across seeds.
>
> > Q5. The previous work [5], which also presents a dexterous manipulation benchmark with musculoskeletal hands, detracts from the novelty of this work.
>
> R5: We agree that the MyoSuite benchmark has been a big step forward to allow musculoskeletal explorations of manipulations. Our work builds ahead. First, in the current work we look beyond single solutions (Experts). We aim to find generalizable behavioral priors that underlines all movements. Second, we extend the MyoHand presented in the MyoSuite by adding more degrees of freedom which allow us to move from in hand rotations of objects to full hand+arm manipulation behaviors. Third, the tasks considered are orders of magnitude more (58, see Table A.3) than the 4 hand rotation tasks presented in MyoSuite. Fourth, the degree of complexity to hold, move and rotate an object is completely different (and extremely more challenging) than the object rotation presented in MyoSuite.
>
> Also see our general response where we explicitly outline our contributions over prior work.
>
>
> Please let us know if our response answers the reviewer’s questions. We are happy to provide additional clarifications to improve our score. We thank the reviewer again for their time and effort helping us improve our paper!

---

### Official Review · Reviewer_RfAu · 2022-10-29

**Confidence:** 3
**Clarity, Quality, Novelty And Reproducibility:** The paper is clear, but the results a…
**Correctness:** 4
**Technical Novelty And Significance:** 2
**Empirical Novelty And Significance:** 2
**Recommendation:** 3

**Strength And Weaknesses:**

Strength:

- the paper constructs an environment for studying dexterous physiological manipulation problems.
- The paper illustrates the potential of reinforcement learning in multi-task learning.

Weakness:

- Though the paper makes an effective empirical study of RL for dexterous manipulation, I wonder if this is a significant contribution given previous progress in learning dexterous manipulation with RL (Chen et al. (2021)). I do not see significant differences between a MyoHand and a ShadowHand for an RL agent.
- For the baseline evaluation, simply running the supervised learning is not enough; it is better to use imitation learning, at least DAPG, to have some RL components to fine-tune the student agent.

**Summary Of The Paper:**

This paper presents a reinforcement learning problem on MyoHand aiming at learning generalizable representation for manipulating different objects to finish different tasks. The authors designed an RL problem based on the previous MyoSim platform and trained agents to solve 14 tasks together. Grasp pose detection is applied to simplify the RL training. Experiments show that training together outperforms the baseline of first training experts separately and then distilling them into one policy through behavior cloning. Authors show that MyoDex, the agent trained to solve all tasks, has a certain level of zero-shot generalization ability and can be fine-tuned on single tasks.

**Summary Of The Review:**

The paper presents a  study of training multi-task RL agents on MyoHand. I appreciate the efforts in designing tasks and running RL experiments. Still, the task of grasping and moving objects with a dexterous hand could be more novel, while the authors only use existing RL methods to solve this task without further improvements. Such contributions do not reach the acceptance bar. I suggest authors figure out challenges in dexterous hand manipulation and design or evaluate algorithms, showing the value of studying such problems.

---

> ### Author Response · Authors · 2022-11-10
> **Point to point response**
>
> We thank the reviewer for their feedback on our work and the acknowledgment of our contributions in dexterous physiological manipulation. We have addressed the reviewer comments in a point-to-point manner below. Please let us know if we can provide any additional clarifications during the discussion period to improve our score.
>
> > Q1: Though the paper makes an effective empirical study of RL for dexterous manipulation, I wonder if this is a significant contribution given previous progress in learning dexterous manipulation with RL (Chen et al. (2021)). I do not see significant differences between a MyoHand and a ShadowHand for an RL agent.
>
> R1: We feel there has been some confusion. Our main contribution is not to show that RL works for dexterous manipulation. Our goal is to acquire a generalizable behavioral prior that can aid in unseen dexterous manipulation tasks (see general comment). Reinforcement learning has made significant progress in dexterous manipulation. Nevertheless, it is quite ineffective due to exploration challenges in high dimensional problems such as biomechanical systems [Schumacher et al 2022] and is limited in generalization between tasks. Our behavioral prior MyoDex overcomes these challenges proving acceleration as well as solving tasks that otherwise can’t be solved.
> More specifically with respect to [Chen et al], here:
>
> 1. We focus on overactuated systems and look for generalizable solutions that can be used as transferable priors to overcome the learning limitation of single/expert solutions.
>
> 2. The generalization offered by MyoHand allows us to learn quickly new tasks with very few iterations [247 iterations to reach 0.8 success rate, see Table A.3] as opposed to extensive explorations needed by Chen et al (more than 1e9 iterations).
>
> 3. We solve full hand manipulations tasks as opposed to only in-hand rotations in  Chen et al. (2021).
>
> > Q2: For the baseline evaluation, simply running the supervised learning is not enough; it is better to use imitation learning, at least DAPG, to have some RL components to fine-tune the student agent.
>
> R: We are unclear about the exact suggestion here. Our baseline is one of the strongest state of the art published methods  (Jain et al. 2019, Chen et al. (2021) which has demonstrated task level generalization. Additionally, DAPG assumes access to expert trajectories of the tasks. We do not assume such access for downstream tasks. The downstream task is specified only using desired object-trajectories. DAPG will require full access to agents' states and actions which are unavailable. We apologize if we didn’t understand and answer the exact question. It will help if the reviewer can rephrase and concretely state the details of this suggestion.
>
>
> Please let us know if our response answers the reviewer’s questions. We are happy to provide additional clarifications to improve our score. We thank the reviewer again for their time and effort helping us improve our paper!

---

### Author Response · Authors · 2022-11-10
**General Comments**

We thank the reviewers for recognizing the contribution of the introduced task set to study physiological dexterous manipulation. However, based on some of the reviewers’ remarks, we believe the primary contributions of our work might have been partially overlooked in light of the features of our task-set and overall effectiveness of MyoDex in solving them. Drawing motivation from biological agents which seldom learn behaviors in isolation – we reiterate – “the primary contribution of the proposed work is to achieve a generalizable and universal behavioral prior that can aid any unseen dexterous manipulation task.”

We are inspired by a few recent works - Pregrasp (Dasari et al 2022) and MyoSuite (Caggiano et al 2022)  - and make following contributions over them:

- Generalization
   - Unlike Pregrasp and MyoSuite which provide single task individual behaviors and showcase no generalization/ transfer to new unseen tasks, primary goal of MyoDex is to generalize via behavioral priors to unseen tasks (38 tasks, Figure 8).

- Pregrasp
  - We emphasize that experts using PreGasp are not able to solve 28 of the tasks we study. On the other hand, MyoDex behavioral priors not only accelerate learning but also solve those 28 tasks for which expert solutions were not found (Figure 8 and Table A.3).

  - Simply distilling the information from PreGasp-experts doesn’t lead to effective behavioral prior that can be leveraged to solve unseen tasks. MyoDex priors on the other hand are highly effective (Distilled experts exhibit 1 shared synergy vs 5 shared with MyoDex, Figure A.6).

  - Fine-tuning Pre-grasp expert solutions while sometimes effective is not enough. Only 2 tasks, amongst 14, saw benefits from fine-tuning Pre-grasp experts (Figure A.8). Whereas all 14 saw benefits from MyoDex (Figure 8).

- MyoSuite
  - Behaviors are restricted to in-hand manipulation. Thanks to MyoDex action priors, we can produce both in-hand and full arm behaviors in 38 unseen tasks.
  - We extend MyoHand with a full arm and embed it in a comprehensive task-set of 58 dexterous manipulations tasks on complex objects.
  - MyoSuite can only solve a very limited set (4 tasks) of single tasks with fixed goals, whereas MyoDex delivers behaviors with a diverse set (52, 1 order of magnitude larger, Figure 8) of unseen objects and tasks.



**Control of MyoHand vs Shadowhand – differences in humans and robotic control**
The primary contributions of our work is agnostic to the details of the embodied agents for which the reusable representations are being acquired. Given the roots of our motivation in human muscle synergy and ease of analysis, we acquire and demonstrate the effectiveness of MyoDex representations on MyoHand. We emphasize that our method can be leveraged for any high dof agent, including Adroit/shadowhand.

In contrast to Adroit, which is actuated using 24 motors, physiological MyoHand is actuated using 39 muscles. The curse of dimensionality due to the large action space has been a challenge in the biomechanic community. The SOTA results are restricted to simple posing and basic grasping behaviors. Thanks to the generalization infused by MyoDex action prior, we demonstrate for the first time a heterogeneous set of contact rich manipulation strategies across a large set (28, see Table A.3) of previously unsolved tasks by experts.

We have uploaded a revisited version of the manuscript based on the review-feedback and highlighted major changes in red.

**Major changes**:

1. We have added 30 additional test-tasks evaluating the effectiveness of our behavioral prior MyoDex (Figure 8 and Table A.3). Results strongly indicate the benefits of leveraging MyoDex in accelerating learning and avoiding local minimas. We also present a few corner cases where expert policies actually perform better.

2. We have also added a control test (Figure A.8) to demonstrate that neither PreGrasp alone, nor fine-tuning expert policies can be effectively leveraged as behavioral priors.

3. We have added experiments over multiple seeds (3 seeds, see Figure 4, A.2, and A.7) demonstrating that our seed to seed variance is very low.

---

### Decision · Program_Chairs · 2023-01-20

**Decision:**

Reject

**Justification For Why Not Higher Score:**

I did not provide a higher score given the concerns of the reviewers, especially the concern about the contribution of the work, use of ground truth object information and lack of use of any object geometry or visual data.

**Justification For Why Not Lower Score:**

N/A

**Metareview: Summary, Strengths And Weaknesses:**

This paper demonstrates musculoskeletal agents (in simulation) exhibiting an ability to learn multiple object manipulation contact rich tasks jointly. This joint learning leads to a policy that can be fine tuned effectively on several downstream (and unseen at training) tasks.

Four reviewers reviewed this paper. One reviewer provided a borderline positive review and three reviewers provided a reject/borderline reject rating.

The reviewers found the contact rich tasks challenging, found the problem of controlling small, dexterous muscle groups understudied, and found the proposed system to be simple and an important baseline for future work. While some concerns of reviewers were addressed well by the rebuttal and new experiments, a few major concerns remained. The primary one was the main contribution of this work, i.e. to show that joint training or multi-task training led to a policy that can be effectively finetuned on more downstream tasks. The reviewers concurred that this contribution was not sufficient for acceptance to this conference. In addition, they pointed out that several other baselines to learn generalizable policies like meta-learning could have been compared to, to strengthen the work. Another concern was the use of the ground truth state information which ignored the geometry of the object and also the lack of using visual inputs to infer any information about the geometry. Such information could lead to a stronger model with more generalization capabilities and also a stronger paper submission breaking new ground in this area. The detailed reviews also outline some other concerns that remained post rebuttal. Given these concerns which I think are valid and important to address, the ratings and my reading of the paper, I recommend a reject.